# Severe maternal morbidity and its associated factors: A cross-sectional study in Morang district, Nepal

Sushma Rajbanshi[1], Mohd Noor Norhayati[2]*, Nik Hussain Nik Hazlina[1]

1 School of Medical Sciences, Women's Health Development Unit, Universiti Sains Malaysia, Kubang Kerian, Malaysia, 2 Department of Family Medicine, School of Medical Sciences, Universiti Sains Malaysia, Kubang Kerian, Malaysia

* hayatikk@usm.my

**Data Availability Statement:** All relevant data are within the paper and its Supporting information files.

## Abstract

### Background

Understanding maternal morbidity and its determinants can help identify opportunities to prevent obstetric complications and improvements for maternal health. This study was conducted to determine the prevalence of severe maternal morbidity (SMM) and the associated factors.

### Methods

A hospital-based cross-sectional study was conducted at Koshi Hospital, Nepal, from January to March 2020. All women who met the inclusion criteria of age ≥18 years of age, Morang residents of Nepalese nationality, had received routine antenatal care, and given birth at Koshi Hospital were recruited consecutively. The World Health Organization criteria were used to identify the women with SMM. A multiple logistic regression analysis was performed. Overall, 346 women were recruited.

### Findings

The prevalence of SMM was 6.6%. Among the SMM cases, the most frequently occurring SMM conditions were hypertensive disorders (12, 56.5%), hemorrhagic disorders (6, 26.1%), and severe management indicators (8, 34.8%). Women with no or primary education (adjusted odds ratio: 0.10, 95% confidence interval: 0.01, 0.76) decreased the odds of SMM compared to secondary education.

### Conclusion

The approximately 7% prevalence of SMM correlated with global studies. Maternal education was significantly associated with SMM. If referral hospitals were aware of the expected prevalence of potentially life-threatening maternal conditions, they could plan to avert future reproductive complications.

**Funding:** This research was funded by the Universiti Sains Malaysia Graduate Development Incentive Grant 311/PPSP/4404808. The funder had no role in the study design, data collection, analysis, decision to publish, or manuscript preparation.

**Competing interests:** The authors have declared that no competing interests exist.

## Introduction

Maternal mortality is a public health problem studied worldwide [1], but the existing research on maternal mortality represents only a fraction of the problem [2]. Globally, the maternal mortality ratio has declined by 38% between 2000 and 2017; the greatest decrease during this period was in Southern Asia, with a nearly 60% reduction in maternal mortality ratio [3]. Maternal near-miss (MNM) and severe maternal morbidity (SMM) are new strategic indicators of maternal health conditions [4]. The World Health Organization (WHO) adopted and defined MNM and SMM standard criteria in 2009 [5]. The purpose of developing these uniform criteria was to provide common ground for comparisons across countries [5, 6]. MNM refers to "a woman who nearly died but survived a complication that occurred during pregnancy, childbirth or within 42 days of termination of pregnancy" [5]. The WHO working group has recommended the use of the term MNM as it best reflects the severity of events. However "severe acute maternal morbidity" (SAMM) is also used for MNM [5]. The WHO uses clinical-, laboratory-, and management-based criteria to identify MNM [7].

If observed across a broad spectrum, women's reproductive health starts from a healthy pregnancy, morbidity, severe morbidity, near miss, and ends at maternal death. SMM lies somewhere between these two spectra before near-miss [8, 9]. Severe maternal morbidities are less in severity than MNM [5]. SMM includes women who did not necessarily have a critical illness but suffered complications related to pregnancy, delivery, and puerperium [5]. The WHO defines SMM as "potentially life-threatening conditions during pregnancy, childbirth, or after the termination of pregnancy from which maternal near-miss cases would emerge" and is assessed based on the four standard conditions, which are (i) hemorrhagic disorders, (ii) hypertensive disorders, (iii) other systemic disorders, and (iv) severe management indicators [5]. The terms "maternal near miss" and "severe maternal morbidity" are used interchangeably in the literature, but SMM reflects a less severe condition than MNM [9, 10]. While both "SMM" and "potentially life-threatening conditions" are used, SMM will be applied in this study.

The extent of MNM has been studied widely [11]; however, limited studies are available on SMM. In Nepal, reported MNM prevalence ranged from 3.8 per 1000 live births in 2013 [12] to a maximum of 23.1 per 1000 deliveries in 2010 [13], while none are reported on SMM. Many studies, especially those in low-income countries, have used a modified version of the WHO near-miss approach, mainly due to its limited applicability in low-income settings, notably due to the laboratory- and management-based criteria [14, 15].

It is necessary to determine a relevant maternal morbidity measurement and investigate its associated factors to improve maternal healthcare services because maternal deaths are becoming rare events [10, 16, 17]. Furthermore, it may be too late for intervention if at-risk women are identified late in labor [18]. Studies on SMM determinants will add valuable information to identify opportunities for prevention and improvements of the quality of obstetric complications at an earlier stage [19]. The purpose of this study was to determine the prevalence of SMM and associated factors. The WHO-based SMM criteria [11] were used in this study.

## Materials and methods

A hospital-based cross-sectional study was conducted at Koshi Hospital, Nepal, from January to March 2020. Morang district was chosen for its dense population, high patient flow, diversified ethnic composition, and mixed population of urban and rural areas. This facility was selected purposefully because Koshi Hospital alone covers more than 90% of the Morang district's total deliveries (i.e., about 9000 deliveries per year) [20]. Koshi Hospital is located in an urban area in the Morang district and is a referral hospital.

The study population comprised women who gave birth in the Morang district, and the source population included all women who gave birth at Koshi Hospital. The eligible participants were women aged ≥18 years, Nepalese citizens residing in Morang district who had received routine antenatal care and had given birth at Koshi Hospital. Women more than 42 days postpartum were excluded from the study. Consecutive sampling was applied to recruit eligible participants based on the birth records at Koshi Hospital.

The sample size was calculated using Power and Sample Size Calculation software version 3.1.6 based on comparing two proportions. The proportion of SMM women without previous cesarean section experience was 13.4% [21], the proportion of SMM women with SMM was taken 28% based on expert opinion. The difference between women with and without SMM with previous cesarean section was estimated at 14.6%. The ratio of non-SMM to SMM was taken as 2:1. 95% confidence interval and 80% statistical power were used. Based on this information, the calculated sample size was 288, 96 respondents of women with SMM and 192 respondents without SMM. After considering a 20% non-response rate, the required sample size was 346.

The case report form (S1 and S2 Files) included sociodemographic information, previous obstetric history, and current obstetric conditions. Categorizations of sociodemographic variables were as follows. Ethnicity was categorized into Brahmin/Chhetri (the advantaged groups), Janajati (indigenous community), Dalits (regarded as untouchables), Muslim, and others (Marwadi), and Terai/Madhesi (native inhabitants of the flat southern region of Nepal) [22]. Religion was categorized into Hindu and Islam/others (Jain) [23]. Wealth quintiles divide the population into five quintiles (lowest, second, middle, fourth, and highest) based on the ownership of assets. The lowest quintile is the poorest population, and the highest quintile is the wealthiest population. In this study, five wealth quintiles were recategorized into lowest/second, middle, highest/fourth [23, 24]. Place of residence is an administrative division based on the population density, previous five years' annual income, and other facilities available in the area [25]. In this study, the place of residence was categorized into the rural municipality and urban municipality. Education was categorized into no formal education/primary (1 to 5 grade), secondary (6 to 10 grade), and tertiary (11 grade and above). Occupation for women was categorized into housewife/agriculture and others (professional/managerial/self-employed) [23]. Occupation for a husband was categorized into professional/managerial/clerical, sales and services, unskilled manual/agriculture, and others [23]. Hospital records were reviewed, and face-to-face interviews were conducted. The SMM criterion was considered fulfilled if it was stated in the medical record. After childbirth, the medical records of women were retrieved retrospectively on the discharge day, to collect information on SMM conditions based on the standard WHO criteria.

The women were recruited consecutively until the required sample size was achieved. The participants were recruited daily at the postpartum ward and cabins when it was confirmed that the women had been discharged. The data were collected by a trained research assistant with an undergraduate nursing certificate supervised by the Principal Investigator (PI), a Nepalese Ph.D. candidate. After ensuring participants' eligibility, the women were approached to enroll in the study and asked for their written informed consent. The participants' sociodemographic characteristics and previous pregnancy history were collected via face-to-face interviews with women in a stable condition on discharge day. The criteria for the WHO SMM were checked from the medical discharge note. SMM was confirmed when a woman had at least one marker among postpartum hemorrhage, severe preeclampsia, eclampsia, sepsis or severe systemic infection, uterine rupture, or when one of the following interventions was performed: the use of blood products, laparotomy, or admission to the intensive care unit. The PI reconfirmed the data on SMM criteria.

The data were cleaned and analyzed using IBM SPSS Statistics version 26.0. The outcome variable was SMM status. The independent variables were sociodemographic variables, previous obstetric history, and current obstetric conditions. A descriptive analysis was used to determine the prevalence of SMM. The numerical variables were presented as means with standard deviations or medians with interquartile ranges. The categorical variables were presented as frequencies and percentages. A simple logistic regression analysis was performed, and all the clinically important variables or variables with p-values ≤0.30 were included in the multiple logistic exploratory regression analysis. Backward and forward methods were employed. Significant variables were analyzed for multicollinearity and interaction, and the Hosmer–Lemeshow goodness of fit test was used. The OR and 95% CI were calculated, and a p-value <0.05 was considered statistically significant.

Ethical approval was obtained from the Human Research Ethics Committee Universiti Sains Malaysia (USM/JEPeM/19060356) and the Nepal Health Research Council (Reg. no. 336/2019). The written consent of the women who agreed to participate in the study was taken before their enrolment. Permission was obtained from the hospital management to review the participants' medical records.

## Results

A total of 346 women were included in the present study. The prevalence of SMM was 6.6%. The most frequently occurring SMM conditions were hypertensive disorders (56.5%) and hemorrhagic disorders (26.1%). Eight (34.8%) women were identified as fulfilling the severe management indicators. One early neonatal death was recorded in this study. The morbidity conditions among the women overlapped, and in total, there were 23 women with SMM (Table 1).

The majority of study participants were housewives (91.0%) from the Hindu religion (90.2%) with secondary or tertiary level education (72.5%). Women in this study belonged to the highest or second-highest wealth quintile (59.8%), and the majority were from Terai/Madhesi ethnicity (48.9%). Participant's husbands also had secondary or tertiary level education (79.5%), their main occupations were unskilled manual/agriculture (42.2%) followed by sales and services (37.3%), and the majority of them were non-smokers (90.2%). The details of sociodemographic and economic characteristics and previous and current obstetric conditions of the women with and without SMM are shown in Table 2. The proportion of births by cesarean section was higher among the SMM than non-SMM women (43.5% vs. 25.7%). Nearly half (47%) of the information for the variable hemoglobin level was missing, so this variable was removed in the subsequent analysis. However, it was worth to be noted that 32% of the participants had mild (24.3%) and moderate (3.8%) anemia.

**Table 1. Morbidity conditions of the women with severe maternal morbidity (n = 23).**

| Characteristics | n (%) |
|---|---|
| Maternal hemorrhagic disorders | *6 (26.1)* |
| Postpartum hemorrhage | 6 (100.0) |
| Maternal hypertensive disorders | *13 (56.5)* |
| Severe hypertension | 11 (84.6) |
| Eclampsia | 2 (15.4) |
| Maternal severe management indicators | *8 (34.8)* |
| Prolonged hospital stays (> 7 postpartum days) | 6 (75.0) |
| Blood transfusion | 2 (25.0) |

**Table 2. Sociodemographic characteristics and previous and current obstetric conditions of the participants with and without severe maternal morbidity (n = 346).**

| Variables | SMM[a] (n = 23) | | non-SMM[a] (n = 323) | |
|---|---|---|---|---|
| | Mean (SD)[b] | n (%) | Mean (SD)[b] | n (%) |
| *Sociodemographic* | | | | |
| Mother's age (year)[c] | 22 (20, 24) | | 22 (20, 25) | |
| Age of marriage (year) | 19.8 (2.23) | | 19.4 (2.38) | |
| Duration of marriage (year)[c] | 1 (1, 4) | | 3 (1, 5) | |
| Ethnicity | | | | |
| Brahmin/Chhetri | | 5 (21.7) | | 34 (10.5) |
| Janajati | | 2 (8.7) | | 49 (15.2) |
| Dalits | | 2 (8.7) | | 51 (15.8) |
| Muslim | | 1 (4.3) | | 33 (10.2) |
| Terai/Madhesi | | 13 (56.5) | | 156 (48.3) |
| Religion | | | | |
| Hindu | | 22 (95.7) | | 290 (89.8) |
| Islam | | 1 (4.3) | | 33 (10.2) |
| Wealth quintile | | | | |
| Lowest/second | | 1 (4.4) | | 19 (5.8) |
| Middle | | 9 (39.1) | | 110 (34.1) |
| Highest/fourth | | 13 (56.5) | | 194 (60.1) |
| Place of residence | | | | |
| Rural Municipality | | 9 (39.1) | | 129 (39.9) |
| Urban Municipality | | 14 (60.9) | | 194 (60.1) |
| Mother's education | | | | |
| No formal education/primary | | 1 (4.3) | | 94 (29.1) |
| Secondary | | 17 (73.9) | | 159 (49.2) |
| Tertiary | | 5 (21.7) | | 70 (21.7) |
| Father's education | | | | |
| No formal education/primary | | 1 (4.3) | | 70 (21.7) |
| Secondary | | 14 (60.9) | | 173 (53.6) |
| Tertiary | | 8 (34.8) | | 80 (24.8) |
| Mother's occupation | | | | |
| Housewife/agriculture | | 21 (91.3) | | 294 (91.0) |
| Professional/managerial/self employed | | 2 (8.7) | | 29 (9.0) |
| Father's occupation | | | | |
| Professional/managerial/clerical | | 2 (8.7) | | 33 (10.2) |
| Sales and services | | 12 (52.2) | | 117 (36.2) |
| Unskilled manual/agriculture | | 8 (34.8) | | 138 (42.8) |
| Others | | 1 (4.3) | | 35 (10.8) |
| Father smoking habit | | | | |
| No | | 19 (82.6) | | 293 (90.7) |
| Yes | | 4 (17.4) | | 30 (9.3) |
| *Previous obstetric history* | | | | |
| Birth spacing (month)[c] | 60 (29.7, 120) | | 36 (24, 60) | |
| Parity | | | | |
| Nullipara | | 17 (73.9) | | 178 (55.1) |
| Multipara | | 6 (26.1) | | 145 (44.9) |
| Previous mode of birth | | | | |
| Nulliparous | | 17 (73.9) | | 178 (55.1) |

(*Continued*)

**Table 2.** (Continued)

| Variables | SMM<sup>a</sup> (n = 23) | | non-SMM<sup>a</sup> (n = 323) | |
|---|---|---|---|---|
| | Mean (SD)<sup>b</sup> | n (%) | Mean (SD)<sup>b</sup> | n (%) |
| Normal birth | | 4 (17.4) | | 123 (38.1) |
| Cesarean section | | 2 (8.7) | | 22 (6.8) |
| History of abortion | | | | |
| No | | 23 (100) | | 310 (96.0) |
| Yes | | 0 (0.0) | | 13 (4.0) |
| *Current obstetric conditions* Period of gestational (week) | 39.0 (1.84) | | 39.0 (1.73) | |
| Number of ANC<sup>d</sup> visits | | | | |
| 4 visits | | 7 (30.4) | | 118 (36.5) |
| ≤3 visits | | 6 (26.1) | | 119 (36.9) |
| ≥5 visits | | 10 (43.5) | | 86 (26.6) |
| Pre-pregnancy BMI<sup>e</sup> (kg/m<sup>2</sup>) | | | | |
| Normal | | 19 (82.6) | | 244 (75.5) |
| Underweight | | 2 (8.7) | | 55 (17.1) |
| Overweight and obese | | 2 (8.7) | | 24 (7.4) |
| Mode of birth | | | | |
| Normal birth | | 13 (56.5) | | 240 (74.3) |
| Cesarean section | | 10 (43.5) | | 83 (25.7) |

Note:

<sup>a</sup> severe maternal morbidity.

<sup>b</sup> standard deviation.

<sup>c</sup> median, interquartile range, skewed towards the right.

<sup>d</sup> antenatal care.

<sup>e</sup> body mass index.

There were 20 independent variables in this study. All the variables were analyzed using simple logistic regression to identify the factors associated with SMM (Table 3). Age of marriage, duration of marriage, mother's education, father's education, parity, previous mode of birth, mode of birth, and the number of ANC visits were the independent variables with $p$-value $<0.3$ that were analyzed in multivariate regression analysis. All independent variables that had shown significant association with SMM were tested for their collinear relationship. None of these variables were found correlated.

The independent variables associated with SMM ($p < 0.05$) determined by multiple logistic exploratory regression analysis are shown in Table 4. The determinant found to be significantly associated with SMM was no formal and primary education (adjusted OR: 0.10, 95% CI: 0.01, 0.76). Women having no formal and primary education decreased the odds of SMM by 9 times than women with secondary education (Table 4). However, among the women with higher education, there was no significant difference in SMM status compared to women with secondary education.

## Discussions

The overall SMM ratio was 66.5/1000 deliveries in the current study. Maternal no or primary education was significantly associated with SMM. The extent of SMM has been reported in limited studies, and they are generally reported together with MNM. A wide range of SMM rates have been reported: the lowest at 1.15% in China in 2013 [26] and the highest at 19.1% in

**Table 3. Factors associated with severe maternal morbidity using simple logistic regression analysis (n = 346).**

| Variables | Crude OR[a] (95% CI[b]) | Wald stat[c] (df)[d] | p-value |
|---|---|---|---|
| *Sociodemographic* | | | |
| Mother's age (year) | 0.96 (0.85, 1.09) | 0.33 (1) | 0.565 |
| Age of marriage (year) | 1.07 (0.94, 1.30) | 1.58 (1) | 0.209 |
| Duration of marriage (year) | 0.92 (0.79, 1.07) | 1.08 (1) | 0.299 |
| Ethnicity | | | |
| Brahmin/Chhetri | 1.76 (0.59, 5.28) | 1.03 (1) | 0.310 |
| Janajati | 0.49 0.10, 2.25) | 0.84 (1) | 0.358 |
| Dalits | 0.47 0.10, 2.15) | 0.94 (1) | 0.332 |
| Muslim | 0.36 0.04, 2.88) | 0.92 (1) | 0.338 |
| Terai/Madhesi | 1 | | |
| Religion | | | |
| Hindu | 1 | | |
| Islam | 0.39 0.05, 3.06) | 0.78 (1) | 0.377 |
| Wealth quintile | | | |
| Lowest/second | 0.78 0.09, 6.33) | 0.05 (1) | 0.821 |
| Middle | 1.22 0.50, 2.94) | 0.19 (1) | 0.657 |
| Highest/fourth | 1 | | |
| Place of residence | | | |
| Rural Municipality | 1.03 0.43, 2.46) | 0.00 (1) | 0.939 |
| Urban Municipality | 1 | | |
| Mother's education | | | |
| No formal education/primary | 0.09 0.01, 0.76) | 4.95 (1) | 0.026 |
| Secondary | 1 | | |
| Tertiary | 0.69 0.24, 1.88) | 0.58 (1) | 0.445 |
| Father's education | | | |
| No formal education/primary | 0.18 0.02, 1.37) | 2.75 (1) | 0.097 |
| Secondary | 1 | | |
| Tertiary | 1.24 0.49, 3.06) | 0.21(1) | 0.648 |
| Mother's occupation | | | |
| Housewife/agriculture | 1 | | |
| Professional/managerial/self-employed | 0.97 0.21, 4.32) | 0.00 (1) | 0.963 |
| Father's occupation | | | |
| Professional/managerial/clerical | 1.04 0.21, 5.15) | 0.00 (1) | 0.956 |
| Unskilled manual/agriculture | 1 | | |
| Sales and services | 1.77 0.70, 4.47) | 1.45 (1) | 0.228 |
| Others | 0.49 0.06, 4.07) | 0.43 (1) | 0.511 |
| Father smoking habit | | | |
| No | 0.48 0.15, 1.52) | 1.53 (1) | 0.216 |
| Yes | 1 | | |
| *Previous obstetric history* | | | |
| Birth spacing (month) | 1.01 0.99, 1.03) | 2.59 (1) | 0.107 |
| Parity | | | |
| Nullipara | 1 | | |
| Multipara | 0.43 0.17, 1.13) | 2.94 (1) | 0.086 |
| Previous mode of birth | | | |
| Normal birth | 1 | | |
| Cesarean section | 2.79 0.48, 16.20) | 1.31 (1) | 0.251 |

*(Continued)*

**Table 3.** (Continued)

| Variables | Crude OR<sup>a</sup> (95% CI<sup>b</sup>) | Wald stat<sup>c</sup> (df)<sup>d</sup> | *p*-value |
|---|---|---|---|
| Nulliparous | 2.93 0.96, 8.94) | 3.59 (1) | 0.058 |
| History of abortion | | | |
| No | 1 | | |
| Yes | 0.00 0.00, 0.00) | 0.00 (1) | 0.999 |
| *Current obstetric conditions* Gestational weeks at birth (week) | 1.01 0.79, 1.29) | 0.00 (1) | 0.923 |
| Number of ANC<sup>e</sup> visits | | | |
| 4 visits | 1 | | |
| ≤3 visits | 0.85 0.28, 2.60) | 0.08 (1) | 0.776 |
| ≥5 visits | 1.96 0.72, 5.35) | 1.72 (1) | 0.189 |
| Pre-pregnancy BMI<sup>f</sup> (kg/m<sup>2</sup>) | | | |
| Normal | 1 | | |
| Underweight | 0.47 0.10, 2.06) | 1.01 (1) | 0.346 |
| Overweight and obese | 1.07 0.23, 4.87) | 0.01 (1) | 0.930 |
| Mode of birth | | | |
| Normal birth | 1 | | |
| Cesarean section | 2.22 0.94, 5.26) | 3.31 (1) | 0.069 |

Note:

<sup>a</sup> odds ratio.

<sup>b</sup> confidence interval.

<sup>c</sup> Wald statistics.

<sup>d</sup> degree of freedom.

<sup>e</sup> antenatal care.

<sup>f</sup> body mass index.

Brazil in 2017 [27]. Studies conducted among high-risk pregnant women and women with type 1 diabetes mellitus in Brazil in 2019 reported the highest rates of SMM at 35.1% [28] and 37.3% [29], respectively. The reported MNM ratios in Nepal were 22.3/1000 deliveries in 2010 [13], 3.8/1000 live births in 2013 [12], and 32.5/1000 deliveries in 2016 [30]. According to the WHO definition, SMM is lower in severity than MNM, so the higher prevalence of SMM in this study is justifiable.

The highest estimates of MNM at 198/1000 live births and 120/1000 live births have been reported in sub-Saharan Africa and the Asia region, respectively [11]. The lowest estimates

**Table 4. Factors associated with severe maternal morbidity using multiple logistic regression analysis (n = 346).**

| Variables | Adj. OR<sup>a</sup> (95% CI<sup>b</sup>) | Wald stat<sup>c</sup> (df<sup>d</sup>) | *p*-value |
|---|---|---|---|
| Maternal education | | | |
| No formal education/primary | 0.10 (0.01, 0.76) | 4.94 (1) | 0.026 |
| Secondary | 1 | | |
| Tertiary | 0.62 (0.21, 1.76) | 0.80 (1) | 0.370 |

Note:

<sup>a</sup> adjusted odds ratio.

<sup>b</sup> confidence interval.

<sup>c</sup> Wald statistics.

<sup>d</sup> degree of freedom.

reported from the Asian region were 4.4/1000 pregnancies [31] and 2.2/1000 live births [32]. The weighted pooled prevalence of MNM worldwide estimated by a systematic review and meta-analysis in 2019 was 18.67/1000 live births [33]. The vast disparities in the prevalence of MNM between high- and low-income nations could be attributed to differences in healthcare systems, particularly maternity care, study populations, and diagnostic criteria and techniques [8]. These disparities can also be explained by an investigation of a single hospital, city, or province, all of which can yield different results within the same country [11].

Maternal hemorrhagic disorders [34–36] and maternal hypertensive disorders [21, 36, 37] are the leading causes of SMM in most studies. Similarly, hypertensive disorders and hemorrhagic disorders were associated with SMM in this study. Interestingly, the literature has repeatedly noticed that hypertensive disorders are more likely to be the leading cause of SMM [10, 35, 37]. In contrast, hemorrhagic disorders are the leading cause of MNM [10, 35, 37]. It suggests that hemorrhage can occur without warning, and delays in managing hemorrhage may lead to maternal death if appropriate obstetric care is not offered, unlike hypertension that is preventable. Lotufo et al. found that hypertension was the leading cause of hospital admissions and potentially life-threatening conditions in their study; however, hemorrhage was the main cause of MNM or even death [35]. Studies have also shown that the case fatality rate of obstetric hemorrhage is higher than that of hypertensive disorders [35, 38]. Hypertensive disorders were found to cause near-miss events but not maternal deaths in a study in Pakistan; however, hemorrhage was responsible for the higher frequency of near-miss events and maternal deaths [39]. In contrast, studies have found case fatality rates higher among women with hypertensive disorders followed by severe postpartum hemorrhage [40, 41]. The higher percentages of both hypertensive disorders and hemorrhagic disorders among women indicate some form of delay in managing obstetric complications by healthcare providers [42].

In the current study, women with secondary or higher education were at 8.9 times at higher odds of developing SMM than women with no or primary education. Similar to this study, maternal higher education was significantly associated with MNM in a case-control study in southeast Iran [43]. The WHO Global Survey on Maternal and Perinatal Health, which was conducted in 24 countries, found that, compared to women with more than 12 years of education, women with no education were at a 2.7-times higher risk and women with 1–6 years of education had twice the risk of maternal mortality [44]. In the secondary data analysis of a demographic health survey in Brazil, women with no or fundamental education had 2.18 times the odds of MNM [45]. The current study findings were not in agreement with most of those in the literature.

Previous studies have shown that utilization of maternal health services increases with higher maternal education [46]. They are more likely to come from a higher socioeconomic situation, have an increased degree of health concern, and have better healthcare access [47, 48]. Women with a higher level of education are more proactive in their maternal health-seeking behavior, such as arranging prenatal checks, seeking professional health care, and likely to choose to give birth in a setting with competent medical staff and well-equipped facilities [48, 49]. In line with this evidence, in the current study, women with higher education who regularly received ANC care from Koshi Hospital identified high-risk factors. They gave birth in the same hospital. Furthermore, a slightly higher percentage of women with high blood pressure (5.2% vs. 1.1%) had a secondary or higher education than lower educated women.

On the contrary, women with no formal education lack access to health information and have no or inadequate ANC visits, which influences their awareness of obstetric complications and access to better medical services [50, 51]. The percentage of homebirths in the Morang district was 38% in the year 2016 [23]. Additionally, women with no or up to primary education

were also more likely to have homebirths and have incomplete ANC. Furthermore, women with a low socioeconomic position and lack of education are more likely to wait until an emergency to seek medical help [52]. Unfortunately, these women were left out of the study, which could be one of the reasons why women with less education have a lower risk of having SMM. Approximately 44% of the women with SMM in this study had undergone a cesarean section.

In contrast, several other studies indicated higher percentages, ranging from 58.4% to 85.8% [21, 26, 35], of women who had given birth via cesarean section among MNM cases. These cesarean sections had been conducted as a required urgent action to prevent complications [37]. Although cesarean section increased the odds of SMM, an association could not be established in this study. Notwithstanding, previous studies have found cesarean section significantly associated with a higher risk of SMM [21, 37].

Studies exploring SMM determinants have found that both nulliparous and multiparous women have the highest risk of SMM [53, 54]. However, parity alone has not been shown to have a consistent association with poor obstetric outcomes [55, 56]. Nulliparous mothers are at an increased risk of hypertension [57] and, if of adolescent age, are not physically capable of childbearing [36], may delay seeking early ANC or birthing services in the event of complications [58, 59], and are in a vulnerable position concerning making decisions for themselves. Grand multipara has been reported as an independent risk factor of gestational diabetes mellitus, antepartum hemorrhage, malpresentation, and postpartum hemorrhage [55, 60].

Studies have shown that parity may be confounded by maternal age [55, 56]. Women of advanced maternal age are likely to have comorbidities [58, 61], which leaves them with less physiological reserve to cope with pregnancy morbidities [62]. Being a mother of advanced age with chronic diseases may influence the gestational prognosis, increasing the chance of complications [10]. Accordingly, parity confounded with younger or advanced age increases the likelihood of SMM. However, similar to this study, studies did not find any significant associations between SMM and maternal age or parity [8, 63]. Although nulliparous women represented 56% of the women in this study, the proportions of women ≤19 and ≥35 years were small, limiting the possibility of drawing further conclusions.

Our study findings have paved the path for referral institutions to begin routine surveillance of SMM cases using WHO criteria derived from standard medical records. Because the standard WHO criteria were followed, our findings are comparable across countries. The current study had limitations that need to be considered, i.e., it was a single hospital-based study; therefore, the findings can be generalized only to the local context in similar demographic and hospital settings. Furthermore, because homebirths, which accounted for 38 percent of births in the Morang district, were not included in the study design, the actual prevalence of SMM is predicted to be greater.

## Conclusions and recommendations

The prevalence of SMM in the current study was in line with that of other studies worldwide. Maternal lower education was associated with SMM. Women who receive routine antenatal checkups should be carefully monitored to, at a minimum, prevent hypertensive and preventable hemorrhage disorders, although postpartum hemorrhage is unpredictable. The study of SMM and its determinants can contribute to formulating strategies to prevent progression to near-miss cases and reduce maternal mortality. The study of potentially life-threatening maternal conditions can reduce future reproductive complications if health institutions know its estimate and are prepared in advance.

## Supporting information

**S1 File. Case report form in English.**
(PDF)

**S2 File. Case report form in Nepali.**
(PDF)

**S3 File. Severe maternal morbidity data set.**
(DTA)

## Acknowledgments

The authors would like to acknowledge Koshi Hospital, staff, and all participants. We are very grateful to all individuals who were, directly and indirectly, involved in this study. We would like to thank Scribendi Inc (www.scribendi.com) for the English Language review.

## Author Contributions

**Conceptualization:** Sushma Rajbanshi, Mohd Noor Norhayati, Nik Hussain Nik Hazlina.

**Data curation:** Sushma Rajbanshi.

**Formal analysis:** Sushma Rajbanshi, Mohd Noor Norhayati.

**Funding acquisition:** Mohd Noor Norhayati.

**Methodology:** Sushma Rajbanshi, Mohd Noor Norhayati, Nik Hussain Nik Hazlina.

**Project administration:** Mohd Noor Norhayati.

**Supervision:** Mohd Noor Norhayati, Nik Hussain Nik Hazlina.

**Visualization:** Mohd Noor Norhayati.

**Writing – original draft:** Sushma Rajbanshi.

**Writing – review & editing:** Sushma Rajbanshi, Mohd Noor Norhayati, Nik Hussain Nik Hazlina.

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
