## [Decision Letter · Decision Letter 0]

10 Sep 2021

PONE-D-21-21068

Severe maternal morbidity and its associated factors: a cross-sectional study in Morang district, Nepal

PLOS ONE

Dear Dr. Nor,

Thank you for submitting your manuscript to PLOS ONE. After careful consideration, we feel that it has merit but does not fully meet PLOS ONE’s publication criteria as it currently stands. Therefore, we invite you to submit a revised version of the manuscript that addresses the points raised during the review process.

We look forward to receiving your revised manuscript.

Kind regards,

Russell Kabir, PhD

Academic Editor

PLOS ONE

2. Please include additional information regarding the survey or questionnaire used in the study and ensure that you have provided sufficient details that others could replicate the analyses. For instance, if you developed a questionnaire as part of this study and it is not under a copyright more restrictive than CC-BY, please include a copy, in both the original language and English, as Supporting Information. If the original language is written in non-Latin characters, for example Amharic, Chinese, or Korean, please use a file format that ensures these characters are visible

3. Please state whether you validated the questionnaire prior to testing on study participants. Please provide details regarding the validation group within the methods section.

“This research was funded by the Universiti Sains Malaysia Graduate Development Incentive Grant 311/PPSP/4404808. The funder had no role in the study design, data collection, analysis, decision to publish, or manuscript preparation.”

Additional Editor Comments (if provided):

Reviewers' comments:

Reviewer's Responses to Questions

**Comments to the Author**

1. Is the manuscript technically sound, and do the data support the conclusions?

Reviewer #1: Yes

Reviewer #2: No

Reviewer #3: Yes

Reviewer #4: Partly

2. Has the statistical analysis been performed appropriately and rigorously? 

Reviewer #1: Yes

Reviewer #2: No

Reviewer #3: Yes

Reviewer #4: No

3. Have the authors made all data underlying the findings in their manuscript fully available?

Reviewer #1: Yes

Reviewer #2: No

Reviewer #3: Yes

Reviewer #4: Yes

4. Is the manuscript presented in an intelligible fashion and written in standard English?

Reviewer #1: Yes

Reviewer #2: Yes

Reviewer #3: Yes

Reviewer #4: No

5. Review Comments to the Author

Reviewer #1: Dear Authors,

I congratulate you on this study, the study has been elaborated nicely and manuscript is understandable the tables are elaborate and self explanatory. the discussion part needs to be build up a little more, especially the part where education was negatively correlated with Severe maternal mortality. The authors need to justify the findings why that is applicable for Nepal settings. the Authors need to write the recommendation of their findings and also propose the utlization of their research findings to make the paper acceptable.

Reviewer #2: PONE-D-21-21068

The authors had made an interesting attempt at exploring SMM and its associated factors among Nepalese women. However, there are serious concerns and flaws that need to be addressed.

I suspect ethical concern for this manuscript with regards to previously published article in PLoS One (Rajbanshi et al., 2020, 15(2), e0244072, PLoS ONE), as the ethical approval number for two different studies are same (USM/JEPeM/19060356) and NHRC (Reg. no. 336/2019). However, the methodology described is different including sample size, its calculation, and data collection and so on. Moreover, there is no mention of anything regarding this in this manuscript. This is an unfortunate issue as it then raises ethical questions.

Moreover, there are other major concerns in the technicality of this manuscript.

1. A single center study cannot generalize the data for an entire district.

2. The selection of study site (Morang district) has not been justified. Moreover, if it’s about total number of deliveries per annum, why the authors did not consider taking Thapathali Maternity Hospital of Kathmandu. Moreover, Morang itself is not the district with highest maternal morbidity and mortality in Nepal.

3. Sample size calculation has been done taking the data from Malaysia. Malaysian and Nepalese context is very different. Despite having data of SMM on similar population, for example, Indian women, authors have overlooked its validation of appropriateness in scientific sample size estimation.

4. The authors have mentioned that data was collected by trained research assistant. But, what was the basis? Who has trained them? Did they have prior similar experience?

5. Why history of abortion been taken as independent variable in Table 3, though it did not have data for category “No”?

6. The authors have not mentioned anything about which variables were included in multiple logistic regressions. Which method did they use? Moreover, it seems they have only introduced variables with p<0.05 in bivariate analysis, which is statistically incorrect.

7. Discussion need to be relooked. The authors need to consider studies in similar contexts such as India, Bangladesh etc to compare and contrast as the context in these countries is more or less same with Nepal.

8. Conclusion is not in line with study objective and findings.

Other minor concerns:

1. No explanation of results of table 3 in text. It’d hinder readers from understanding the findings clearly.

2. The logic behind clubbing secondary education with tertiary is not justifiable. Moreover, these level of education need to be defined contextually from a larger readers’ perspective.

3. It’s not enough to mention “no formal education….decreased the odds of SMM” in the abstract. Need to interpret the meaning of the value from a larger readers’ perspective.

4. The concluding remarks in the abstract contrasts with the findings presented in the abstract itself. “no formal education….decreased the odds of SMM” vs women with higher education more likely to utilize hospital referral….”. How come the authors have concluded that “birthing practice of women with lower education at the well-equipped hospital should increase”, which is in no line with the study objectives and the findings.

5. Why had the authors clubbed “Newar” with “Brahmin and Chhetri” in the Ethnicity, despite the fact that “Newar” itself comes under “Janajati”? This is not valid and logical.

6. What did the authors mean by professional category in occupation? They need to define it. Also, they need to justify regarding clubbing of professional with clerical under father’s occupation while analyzing the data to explore associations.

7. Authors need to check the interpretation of their results of table 4 in the text “…decreased odds by 0.11 times…” ?????

8. Check Mesh terms for keywords.

Reviewer #3: Dear author,

The research article seems excellent and wonderful. It would have been better if proper correction on introduction of abstract with proper word and clear language will be use. Other seems brilliant.

Reviewer #4: Dear authors ,

There are some comments on your manuscripts:

1) Line 34: Is "WHO criteria" an appropriate Keyword? please find other important key word rather than this.

2) Line 88-93:

a) You are aiming to calculate hospital based prevalence of SMM. But in sample size calculation you are using two proportion sample size calculation with the two group (with and without SMM). I think there is miss-match between your objective and sample size calculation. Could you please clarify this.

b) What is the basis for "The difference between women with and without SMM with previous cesarean section was estimated at 14.6%" sentence?

c) Specify power and confidence level that you have used for sample size calculation

d) As you have two proportion situation for sample size calculation, please specify sample size in each group.

3. Line 97 and 99: Please specify what others are in ethnicity and religion

4. Line 134. On what basis you have set cut off of p-value <0.3 for the selection of variables for multivariate logistic regression model.

5. What is the difference in the outcome of maternal education vs SMM in tabel3 and table 4. In table 4 you told it as a multivariate regression model but there is only one variables. Where are other variables ? Where are the variables that has p-value <0.3 in the univariate model? Please specify which variables have you adjusted to find the result on table 4.

6. You have checked for collinearity but you have not specified about this in the result section. Could you please specify that in the result section too.

7. You have mentioned that you used consecutive sampling. In a year there seems to be 9000 cases in the hospital. In your study duration it is around 2300 cases (rough). There might have been declined cases, incomplete cases etc during data collection. You have started with 346 study participants but Please specify all declined percentage along with the reasons for decline, non-response proportion, etc in the result section. This would be helpful to determine generalizability of the study findings.

8. You have mentioned limitations in the last paragraph of discussion. Please add strengths of your study

6. PLOS authors have the option to publish the peer review history of their article (what does this mean?). If published, this will include your full peer review and any attached files.

Reviewer #1: **Yes: **DEBLINA ROY

Reviewer #2: No

Reviewer #3: No

Reviewer #4: **Yes: **Bikram Adhikari

---

## [Author Response · Author response to Decision Letter 0]

15 Oct 2021

Reviewer #1: 

Dear Authors,

I congratulate you on this study, the study has been elaborated nicely and manuscript is understandable the tables are elaborate and self-explanatory. The discussion part needs to be build up a little more, especially the part where education was negatively correlated with Severe maternal mortality. The authors need to justify the findings why that is applicable for Nepal settings. the Authors need to write the recommendation of their findings and also propose the utlization of their research findings to make the paper acceptable.

Authors response: In the abstract section, the text below was added.

“If referral hospitals were aware of the expected prevalence of potentially life-threatening maternal conditions, they can plan to avert future reproductive complications.”

The discussion section has been elaborated further as below

“Previous studies have shown, utilization of maternal health services increase with higher maternal education [46] as they are more likely to come from a higher socioeconomic situation, had an increased degree of health concerned, and had better access to health care [47, 48]. Women with a higher level of education are more proactive in their maternal health-seeking behavior, such as arranging prenatal checks, seeking professional health care, and likely to choose to give birth in a setting with competent medical staﬀ and well-equipped facilities [48, 49]. In line with this evidence, in the current study, women with higher education who regularly received ANC care from Koshi Hospital identified with high-risk factors, they gave birth in the same hospital. Furthermore, a slightly higher percentage of women with high blood pressure (5.2% vs. 1.1%) had a secondary or higher education than lower educated women.

On the contrary, women with no formal education lack access to health information and have no or inadequate ANC visits, which inﬂuences their awareness of obstetric complications and access to better medical services [50, 51]. The percentage of homebirths in the Morang district was 38% in the year 2016 [23]. Additionally, women with no or up to primary education were also more likely to have homebirths and have incomplete ANC. Furthermore, women with a low socioeconomic position and lack education are more likely to wait until an emergency to seek medical help [52]. Unfortunately, these women were left out of the study, which could be one of the reasons why women with less education have a lower risk of having SMM.” (Pgs. 16-17, lines 26-286)

Reviewer #2: PONE-D-21-21068

The authors had made an interesting attempt at exploring SMM and its associated factors among Nepalese women. However, there are serious concerns and flaws that need to be addressed.

I suspect ethical concern for this manuscript with regards to previously published article in PLoS One (Rajbanshi et al., 2020, 15(2), e0244072, PLoS ONE), as the ethical approval number for two different studies are same (USM/JEPeM/19060356) and NHRC (Reg. no. 336/2019). However, the methodology described is different including sample size, its calculation, and data collection and so on. Moreover, there is no mention of anything regarding this in this manuscript. This is an unfortunate issue as it then raises ethical questions.

Authors response: Thank you. The previously submitted manuscript and the current manuscript submitted to PLoS One are part of the mixed-method study by the author Rajbanshi, S. for her Doctor of Philosophy thesis at Universiti Sains Malaysia (USM). Ethical approval was taken for this research from USM ethical board and Nepal Health Research Council, which has been mentioned in all submitted manuscripts for publications, in this and other journals. The current manuscript is of a different study design i.e. cross-sectional study. It was one of the phases involved in the research. 

Moreover, there are other major concerns in the technicality of this manuscript.

1. A single center study cannot generalize the data for an entire district.

Authors response: Corrections made in the strengths and limitation sections as below. 

“The current study had limitations that need to be taken into consideration, i.e., it was a single hospital-based study; therefore, the ﬁndings can be generalized only to the local context in similar demographic and hospital settings.” (Pg. 18, lines 314-318)

2. The selection of study site (Morang district) has not been justified. Moreover, if it’s about total number of deliveries per annum, why the authors did not consider taking Thapathali Maternity Hospital of Kathmandu. Moreover, Morang itself is not the district with highest maternal morbidity and mortality in Nepal.

Authors response: Corrections made in the materials and methods sections as below.

“Morang district was chosen for its dense population, high patient flow, diversified ethnic composition, and mixed population of urban and rural areas.” (Pg. 5, lines 88-89).

It was mentioned above that this study was part of a bigger study that required a mixed population of urban and rural areas where other aspects of maternal health were explored; therefore, although Thapathali Maternity Hospital had the highest number of births, Morang district was purposively selected.

3. Sample size calculation has been done taking the data from Malaysia. Malaysian and Nepalese context is very different. Despite having data of SMM on similar population, for example, Indian women, authors have overlooked its validation of appropriateness in scientific sample size estimation.

Authors response: Thank you for your comment. There were numerous studies on maternal near miss, which can be referred to in this study for sample size calculation. This research has utilized a study by Norhayati (2016) because it is recent and applies the same defining WHO criteria for severe maternal morbidity for its outcome. 

4. The authors have mentioned that data was collected by trained research assistant. But, what was the basis? Who has trained them? Did they have prior similar experience?

Authors response: The research assistants were recently graduate nursing from Koshi Hospital. They did not have any experience in research and the utilization of the WHO criteria. They were trained by the Principal Researcher and monitored daily. 

5. Why history of abortion been taken as independent variable in Table 3, though it did not have data for category “No”?

Authors response: During the literature review, few studies had shown that history of abortion was associated with maternal near miss and also severe maternal morbidity. Therefore, it was taken as one of the possible determinants for SMM in this study. 

6. The authors have not mentioned anything about which variables were included in multiple logistic regressions. Which method did they use? Moreover, it seems they have only introduced variables with p<0.05 in bivariate analysis, which is statistically incorrect.

Authors response: The authors mentioned in the manuscript that those variables that had p-values lesser than 0.3 and clinically important variables were forwarded to multiple logistic regressions. The analysis process was mentioned clearly in the methods section as below. We have also added a sentence to clarify further the variables included.

“A simple logistic regression analysis was performed, and all the clinically important variables or variables with p-values ≤0.30 were included in the multiple logistic exploratory regression analysis. Backward and forward methods were employed. Significant variables were analyzed for multicollinearity and interaction, and the Hosmer–Lemeshow goodness of fit test was used. The OR and 95% CI were calculated, and a p-value <0.05 was considered statistically significant.” (Pg. 8, lines 148-153)

7. Discussion need to be relooked. The authors need to consider studies in similar contexts such as India, Bangladesh etc to compare and contrast as the context in these countries is more or less same with Nepal.

Authors response: The text below was added in the discussion section to make the findings comparable in a wider context.

“The reported MNM ratios in Nepal were 22.3/1000 deliveries in 2010 [13], 3.8/1000 live births in 2013 [12], and 32.5/1000 deliveries in 2016 [30]. According to the WHO definition, SMM is lower in severity than MNM, so the higher prevalence of SMM in this study is justifiable.” (Pg. 14, lines 216-219)

“The highest estimates of MNM at 198/1000 live births and 120/1000 live births have been reported in sub-Saharan Africa and the Asia region, respectively [11]. The lowest estimates reported from the Asian regions were 4.4/1000 pregnancies [31] and 2.2/1000 live births [32]. The weighted pooled prevalence of MNM worldwide estimated by a systematic review and meta-analysis in 2019 was 18.67/1000 live births [33]. The vast disparities in the prevalence of MNM between high- and low-income nations could be attributed to differences in healthcare systems, particularly maternity care, study populations, and diagnostic criteria and techniques [8]. These disparities can also be explained by an investigation of a single hospital, city, or province, all of which can yield different results within the same country [11].” (Pg. 14, lines 223-232)

8. Conclusion is not in line with study objective and findings.

Authors response: Conclusions in the abstract and the conclusion section was rewritten as below.

Abstract: “The approximately 7% prevalence of SMM correlated with global studies. Maternal education was significantly associated with SMM. If referral hospitals were aware of the expected prevalence of potentially life-threatening maternal conditions, they could plan to avert future reproductive complications.” (Pg. 2, lines 35-41)

Conclusion section: “The study of potentially life-threatening maternal conditions can reduce future reproductive complications if health institutions know its estimate and are prepared in advance.” (Pg. 18, lines 328-332)

Other minor concerns:

1. No explanation of results of table 3 in text. It’d hinder readers from understanding the findings clearly.

Authors response: The text below was added just above Table 3.

“Age of marriage, duration of marriage, mother’s education, father’s education, parity, previous mode of birth, mode of birth, and the number of ANC visits were the independent variables with p-value <0.3 that were analyzed in multivariate regression analysis.” (Pg. 11, line 187-190)

2. The logic behind clubbing secondary education with tertiary is not justifiable. Moreover, these level of education need to be defined contextually from a larger readers’ perspective.

Authors response: We have redefined the education variable and made necessary changes in Tables 2, 3, and 4. We have also included an explanation of the term in the methods and materials section as below.

 “Education was categorized into no formal education/primary (1 to 5 grade), secondary (6 to 10 grade), and tertiary (11 grade and above).” (Pg. 7, lines 120-122)

3. It’s not enough to mention “no formal education….decreased the odds of SMM” in the abstract. Need to interpret the meaning of the value from a larger readers’ perspective.

Authors response: Corrections made as below

“Women having no formal and primary education decreased the odds of SMM by nine times than women with secondary education (Table 4). However, among the women with higher education, there was no significant difference in SMM status compared to women with secondary education.” (Pg. 13, lines 205-208)

The explanation of the term education variable was added in the methods and materials section as below.

“Education was categorized into no formal education/primary (1 to 5 grade), secondary (6 to 10 grade), and tertiary ( 11 grade and above).” (Pg. 7, lines 120-122)

4. The concluding remarks in the abstract contrasts with the findings presented in the abstract itself. “no formal education….decreased the odds of SMM” vs women with higher education more likely to utilize hospital referral….”. How come the authors have concluded that “birthing practice of women with lower education at the well-equipped hospital should increase”, which is in no line with the study objectives and the findings.

Authors response: Thank you for your comments. The conclusion was rewritten as below.

Abstract: “The approximately 7% prevalence of SMM correlated with global studies. Maternal education was significantly associated with SMM. If referral hospitals were aware of the expected prevalence of potentially life-threatening maternal conditions, they could plan to avert future reproductive complications.” (Pg. 2, lines 35-41)

Conclusion section: “The study of potentially life-threatening maternal conditions can reduce future reproductive complications if health institutions know its estimate and are prepared in advance.” (Pg. 18, lines 328-332)

5. Why had the authors clubbed “Newar” with “Brahmin and Chhetri” in the Ethnicity, despite the fact that “Newar” itself comes under “Janajati”? This is not valid and logical.

Authors response: Thank you for your critical review. The Newar ethnicity was combined together with Janajati, and reanalysis was done. Changes were made in Table 2 and 3.

6. What did the authors mean by professional category in occupation? They need to define it. Also, they need to justify regarding clubbing of professional with clerical under father’s occupation while analyzing the data to explore associations.

Authors response: The authors had taken the reference from the Demographic and Health Survey, which was also mentioned in the manuscript. The Professional category is very broad to mention in the manuscript and interested audience can refer to the reference. The reasons for grouping the professional and clerical occupations were that there were only three that belonged to clerical occupation. It cannot be grouped with unskilled manual as clerical were table work job; therefore it was grouped with the professional category. 

7. Authors need to check the interpretation of their results of table 4 in the text “…decreased odds by 0.11 times…” ?????

Authors response: The interpretation was corrected as below.

“Women having no formal and primary education decreased the odds of SMM by nine times than women with secondary education (Table 4). However, among the women with higher education, there was no significant difference in SMM status compared to women with secondary education.” (Pg. 3, lines 205-208)

8. Check Mesh terms for keywords.

Authors response: The keyword “WHO criteria” was removed and replaced with “maternal health”.

Reviewer #3: Dear author,

The research article seems excellent and wonderful. It would have been better if proper correction on introduction of abstract with proper word and clear language will be use. Other seems brilliant.

Authors response: Thank you for your comments. The conclusion was rewritten as below.

Abstract: “The approximately 7% prevalence of SMM correlated with global studies. Maternal education was significantly associated with SMM. If referral hospitals were aware of the expected prevalence of potentially life-threatening maternal conditions, they could plan to avert future reproductive complications.” (Pg. 2, lines 35-41)

Conclusion section: “The study of potentially life-threatening maternal conditions can reduce future reproductive complications if health institutions know its estimate and are prepared in advance.” (Pg. 18, lines 328-332)

Reviewer #4: Dear authors ,

There are some comments on your manuscripts:

1) Line 34: Is "WHO criteria" an appropriate Keyword? please find other important key word rather than this.

Authors response: Thank you for your suggestion. 

The keyword “WHO criteria” was removed and replaced with “maternal health”.

2) Line 88-93:

a) You are aiming to calculate hospital based prevalence of SMM. But in sample size calculation you are using two proportion sample size calculation with the two group (with and without SMM). I think there is miss-match between your objective and sample size calculation. Could you please clarify this.

Authors response: The study aimed to calculate both the prevalence of SMM and also its associated factors. Sample size calculated using a single proportion for SMM using the prevalence of SMM of 17.5% (Pacheco et al., 2014) was 267. It was smaller than the sample size calculated for the associated factors for SMM of 346. Therefore, comparing two proportion sample size calculation for the associated factors was utilized and shown in this manuscript.

b) What is the basis for "The difference between women with and without SMM with previous cesarean section was estimated at 14.6%" sentence?

Authors response: The text below was added to the manuscript.

“The proportion of SMM women without previous cesarean section experience was 13.4% [21], the proportion of SMM women with SMM was taken 28% based on expert opinion.” (Pg. 6, lines 100-101)

c) Specify power and confidence level that you have used for sample size calculation

Authors response: The text below was added to the manuscript.

“95% confidence interval and 80% statistical power was used.” (Pg. 6, line 104)

d) As you have two proportion situation for sample size calculation, please specify sample size in each group.

Authors response: Below line was added to the manuscript in the materials and methods section

“Based on this information, the calculated sample size was 288, 96 respondents of women with SMM and 192 respondents without SMM.” (Pg. 5, line 104-106)

3. Line 97 and 99: Please specify what others are in ethnicity and religion

Authors response: “Others” in ethnicity and religion were specified as Marwadi and Jain within the manuscript.

4. Line 134. On what basis you have set cut off of p-value <0.3 for the selection of variables for multivariate logistic regression model.

Authors response: The cut off of p-value of <0.3 was taken for screening purposes for selection of variables from simple to multiple logistic regression. It is aimed to obtain more variables for multiple logistic regression analysis. In multiple logistic regression analysis, the p-value of <0.05 is still taken to indicate the significance level.

5. What is the difference in the outcome of maternal education vs SMM in tabel3 and table 4. In table 4 you told it as a multivariate regression model but there is only one variables. Where are other variables ? Where are the variables that has p-value <0.3 in the univariate model? Please specify which variables have you adjusted to find the result on table 4.

Authors response: All the variables with p-value <0.3 were listed in the last column of Table 3. It is now added as texts just above Table 3 as well. 

“Age of marriage, duration of marriage, mother’s education, father’s education, parity, previous mode of birth, mode of birth, and number of ANC visits were the independent variables with p-value <0.3 that were analyzed in multivariate regression analysis.” (Pg. 11, lines 187-190)

6. You have checked for collinearity but you have not specified about this in the result section. Could you please specify that in the result section too.

Authors response: Below lines were added to the result section

“All independent variables that had shown significant association with SMM were tested for their collinear relationship. None of these variables were found correlated.” (Pg. 11, lines 190-192)

7. You have mentioned that you used consecutive sampling. In a year there seems to be 9000 cases in the hospital. In your study duration it is around 2300 cases (rough). There might have been declined cases, incomplete cases etc during data collection. You have started with 346 study participants but Please specify all declined percentage along with the reasons for decline, non-response proportion, etc in the result section. This would be helpful to determine generalizability of the study findings.

Authors response: Those respondents listed in the discharge sheets and fulfill the eligibility criteria were approached for their consent for participation. This includes respondents of Morang district residents. All the eligible respondents approached agreed to participation. 

8. You have mentioned limitations in the last paragraph of discussion. Please add strengths of your study

Authors response: The text below was added to the paragraph where limitations was mentioned.

“Our study findings have paved the path for referral institutions to begin routine surveillance of SMM cases using WHO criteria derived from standard medical records.” (Pg. 18, line 312-313)

Authors response: Manuscript contents were changed according to PLOS ONE’s style requirements.

2. Please include additional information regarding the survey or questionnaire used in the study and ensure that you have provided sufficient details that others could replicate the analyses. For instance, if you developed a questionnaire as part of this study and it is not under a copyright more restrictive than CC-BY, please include a copy, in both the original language and English, as Supporting Information. If the original language is written in non-Latin characters, for example Amharic, Chinese, or Korean, please use a file format that ensures these characters are visible

Authors response: The Case Reporting Form both in English and Nepali language are submitted.

3. Please state whether you validated the questionnaire prior to testing on study participants. Please provide details regarding the validation group within the methods section.

Authors response: The Case Reporting Form (CRF) used in this study was burrowed from the World Health Organization SMM criteria. There were few questions regarding the demographic information of the study participants. The CRF sociodemographic information section was translated into the Nepali language. This section was pre-tested among 15 women from the same hospital to test for logical flow, content, and language comprehensibility. To identify the SMM cases, the hospital medical reports had to be referred to; therefore, they were not translated and used in English. Face-to-face training was provided to the research assistant by the Principal Investigator.

Authors response: The correct grant number is 311/PPSP/4404808 

“This research was funded by the Universiti Sains Malaysia Graduate Development Incentive Grant 311/PPSP/4404808. The funder had no role in the study design, data collection, analysis, decision to publish, or manuscript preparation.”

Authors response: The funding-related text is removed from the manuscript. Please include in the Funding Statement “This research was funded by the Universiti Sains Malaysia Graduate Development Incentive Grant 311/PPSP/4404808. The funder had no role in the study design, data collection, analysis, decision to publish, or manuscript preparation.”

---

## [Editor Report · Decision Letter 1]

23 Nov 2021

Severe maternal morbidity and its associated factors: a cross-sectional study in Morang district, Nepal

PONE-D-21-21068R1

Dear Dr. Noor,

We’re pleased to inform you that your manuscript has been judged scientifically suitable for publication and will be formally accepted for publication once it meets all outstanding technical requirements.

Kind regards,

Russell Kabir, PhD

Academic Editor

PLOS ONE
---

## [Editor Report · Acceptance letter]

20 Dec 2021

PONE-D-21-21068R1 

Severe maternal morbidity and its associated factors: a cross-sectional study in Morang district, Nepal 

Dear Dr. Norhayati:

I'm pleased to inform you that your manuscript has been deemed suitable for publication in PLOS ONE. Congratulations! Your manuscript is now with our production department. 

Kind regards, 

on behalf of

Dr. Russell Kabir 

Academic Editor

PLOS ONE